# Comprehensive Assessment of Flow and Other Analytical Methods Dedicated to the Determination of Zinc in Water

**DOI:** 10.3390/molecules26133914

**Published:** 2021-06-26

**Authors:** Paweł Kościelniak, Paweł Mateusz Nowak, Joanna Kozak, Marcin Wieczorek

**Affiliations:** Department of Analytical Chemistry, Faculty of Chemistry, Jagiellonian University, Gronostajowa 2, 30-387 Kraków, Poland; pm.nowak@uj.edu.pl (P.M.N.); j.kozak@uj.edu.pl (J.K.); marcin.wieczorek@uj.edu.pl (M.W.)

**Keywords:** flow analysis, green analytical chemistry, method assessment, RGB model, validation, zinc determination in waters

## Abstract

An original strategy to evaluate analytical procedures is proposed and applied to verify if the flow-based methods, generally favorable in terms of green chemistry, are competitive when their evaluation also relies on other criteria. To this end, eight methods for the determination of zinc in waters, including four flow-based ones, were compared and the Red–Green–Blue (RGB) model was exploited. This model takes into account several features related to the general quality of an analytical method, namely, its analytical efficiency, compliance with the green analytical chemistry, as well as practical and economic usefulness. Amongst the investigated methods, the best was the flow-based spectrofluorimetric one, and a negative example was that one involving a flow module, ICP ionization and MS detection, which was very good in analytical terms, but worse in relation to other aspects, which significantly limits its overall potential. Good assessments were also noted for non-flow electrochemical methods, which attract attention with a high degree of balance of features and, therefore, high versatility. The original attempt to confront several worldwide accepted analytical strategies, although to some extent subjective and with limitations, provides interesting information and indications, establishing a novel direction towards the development and evaluation of analytical methods.

## 1. Introduction

Choosing the best analytical method in general terms from the many procedures available today is not straightforward, especially when it comes to the wide variety of methods for the determination of metals in water. In addition to meeting the requirements relating to validation criteria (accuracy, precision, limit of quantification and others), it is important to ensure high practical and economic efficiency. Criteria such as cost, time, ease of use and other practical requirements are extremely important for practitioners who decide to choose a specific procedure from among the many alternatives available in the literature. The assessment of the method in this respect is made difficult by the fact that in the scientific literature these parameters are often discussed very briefly and without comparison with other alternative analytical approaches.

Another important issue nowadays is the assessment of compliance with the idea of minimizing human impact on the environment and increasing the safety of use, i.e., greenness of the analytical method. The foundations of green analytical chemistry (GAC) developed over the years, starting from the work of Anastas [1], through the later articles by Koel and Kaljurand [2], Armenta et al. [3] and Gałuszka et al. [4]. The last one formulates 12 GAC rules, which postulate, among others, minimizing the consumption of reagents and energy, reducing waste production, increasing user safety and increasing the integration and automation of analytical procedures.

By taking into account the above guidelines, several tools for assessing the greenness of a method such as the National Environmental Methods Index [5], Green Analytical Procedure Index [6], Eco-Scale [7] and Analytical Greenness Calculator [8] have been recently proposed. These tools allow one to graphically express the greenness of a method, using either specially designed or easy to interpret pictograms, using quantitative indicators or in combination. However, they have one common limitation—they do not treat the green and other crucial criteria in the same way, and thus do not express the global method quality.

One of the tools that enable the method to be assessed taking into account analytical, green and practical criteria is the Red-Green-Blue (RGB) model [9], which refers to the known pattern of mixing light with different colors. The model assigns three primary colors—red, green and blue—to three different attributes of the method: analytical, ecological and practical, respectively. The ideal method according to the RGB model is white, which is fully complete, just as white light contains all three primary colors. The RGB model has already been successfully used to evaluate and compare both procedures [10] and analytical methods [11], and its effectiveness has been demonstrated against other models of similar purpose [12]. It also became an inspiration for the recently proposed new idea “White Analytical Chemistry” (WAC) promoting the coherence and synergy of the analytical, ecological and practical attributes, striving for sustainability of analytical methods [13].

In this work, the RGB model was used to compare and evaluate different analytical methods for the determination of the same analyte in the same type of samples. As an example, the methods for determining zinc in water and food were considered. This choice was made due to the great importance of zinc as one of the important factors responsible for biological functions and the state of human health. Its excess in drinking water may be harmful, hence the high accuracy of its determination is desirable. It is also important that elemental water analysis is performed commonly, routinely and on a large scale, covering a very wide range of analytes (including zinc) in various types of samples. Appropriate analytical methods should therefore best meet particularly high requirements in terms of features such as precision, limit of quantification, speed, cost-effectiveness and simplicity of the analytical procedure.

Eight spectrometric and electrochemical analytical methods were selected for the study, differing from each other in terms of the sample preparation procedure and the type of detection system [14,15,16,17,18,19,20,21]. In four methods, the samples were processed and delivered to the detecting unit using dedicated flow systems [14,19,20,21]. Flow techniques, thanks to their natural properties, obviously favor the GAC rules, including increasing work safety, reducing the consumption of reagents and waste production, as well as allowing easy adaptation of green sample processing modules and energy-saving measuring instruments [22,23]. Thanks to the possibility of full automation of the analytical procedure, they also contribute to increasing the precision of analytical results and to reducing the time and costs of a single analysis. Therefore, the attempt to verify these opinions through a comprehensive comparison of flow-based and batch procedures in the group of methods intended for the implementation of the same analytical task was considered to be a particularly interesting issue in this publication. The novelty of this work results from the use of an original approach to method evaluation, and an attempt to evaluate flow methods on a global basis, going beyond the validation parameters as well as green chemistry.

## 2. Materials and Methods

### 2.1. Analytical Methods

Eight methods considered here are presented in Table 1.

Instrumental operation conditions and materials of the SIA-CE/DAD, SP, DPASV, FIA-DAD, FIA-SF, FIA-ICP/MS methods were exactly described in [14,17,18,19,20,21]. The evaluation of the FAAS and ICP/OES methods was performed in our laboratory on the basis of the procedures dedicated to the determination of zinc in natural water [15,16] under the following instrumental conditions: FAAS spectrometer PinAAcle 900 AA (Perkin Elmer, Inc, Shelton, CT, USA) equipped with hollow cathode lamp (Beijing Vacuum Electronics Research Institute, Beijing, China) with operating current 9.0 mA, wavelength 213.86 nm, slit/width of 2.7/1.8 nm, sample aspiration rate 5.0 mL/min and acetylene and air flow 2.5 and 10.0 L/min, respectively; ICP/OES spectrometer Optima 2100 DV (Perkin Elmer, Inc, USA) working with 1300 W plasma power and with plasma, auxiliary and nebulizing gas flows of 15.0, 0.2 and 0.80 L/min, peaks for wavelength 206,200 nm in axial plasma view mode were measured for sample flow rate 1.50 mL/min.

### 2.2. RGB Model

The RGB model used here refers to the 12 general rules divided into red (R1–R4), green (G1–G4) and blue (B1–B4) corresponding to the three attributes of a method: analytical, ecological and practical, respectively [13]. Overall average score, the so-called whiteness, is a measure of the overall quality and sustainability of the method. It is expressed quantitatively, as are its three main components (redness, greenness and blueness), which makes it possible to easily compare methods and indicate better methods in a general sense and in individual areas (colors). The individual assessment criteria are: R1—scope of application (assessed collectively taking into account the number of simultaneously determined analytes, linearity range, variety of sample matrices, resistance to interference, selectivity and robustness/ruggedness), R2—LOD and LOQ values, R3—precision (either direct, intermediate or both), R4—accuracy (relative error or recovery when no other accuracy data are given), G1—reagent toxicity (measured by the number of all pictograms characterizing all reagents used), G2—quantity of reagents and waste (measured by estimating either the total volume or the mass of reagents used, produced waste, or in combination), G3—energy consumption (estimated total consumption of electricity and other utilities), G4—direct impact (impact on humans—additional workplace hazards to the user, impact on the naturalness of nature—use of animals or genetic manipulation/GMO), B1—cost effectiveness (total estimated cost of the analysis taking into account all stages of procedures and equipment—taking into account the effect of depreciation proportional to the time of use of the instrument), B2—time consumption (total estimated time of the analysis taking into account all stages of the analytical procedure), B3—requirements (minimum sample volume for analysis, practical requirements for infrastructure and personnel qualification level), and B4—ease of use (assessed overall on the basis of subjective assessment of the degree of miniaturization, automation/integration and portability of the instruments).

The assessment of these criteria is made by granting a score in the range from 0 to 100. A score of zero means a total lack of acceptance of a given criterion, the worst possible result, while 100 means full satisfaction, a result fully adequate to the planned application of the method. The assessment is made as critically and objectively as possible, based on validation parameters and reliable estimates for costs, analysis time, waste volume, energy consumption and similar parameters. In some cases, such as ease of use, the assessment is necessarily carried out on the basis of available knowledge and intuition. It is also possible to evaluate individual criteria based on predefined guidelines, but in this case such additional standards were not applied. Indicators such as redness, greenness and blueness are calculated as the arithmetic mean of notes for a given attribute, where each awarded note is calculated with the same weight. Whiteness is the average score for all 12 criteria. Different significance of individual criteria for the whole assessment, taking into account the specificity of the method application, is reflected in the level of criticism and requirements for individual criteria, those of less importance, e.g., the amount of sample needed for analysis in the case of readily available water, are assessed less critically, which to some extent, replaces assigning weights to certain aspects. The values of key validation criteria and other parameters that could be objectively estimated on the basis of given analytical procedures are given for each method in Table 2.

### 2.3. Evaluation Step

12 people with extensive professional experience, specializing in various areas of analytical chemistry, participated in the process of evaluating analytical methods. They made the assessment independently of each other without mutual contact. Each evaluator was based primarily on the same values of features characterizing the methods available in the literature (Table 2). Where relevant data were not available, the missing values were subjectively estimated based on the best evaluator’s knowledge and intuition. Everyone approached it with the utmost care and criticism. The basis for the comparison of methods were the average scores assigned to individual criteria, which, combined with a large group of evaluators, allowed significant reduction in subjectivism, and enabled a multi-dimensional analysis of individual methods using the RGB model.

## 3. Results and Discussion

### 3.1. Initial Assessment

At the beginning, the variability of the position of a given method in the ranking, depending on the evaluator, was compared. The ranking was made on the basis of the overall evaluation of the methods expressed by the whiteness parameter, assigning each of them numbers from 1 (the best method) to 8 (the worst method). The collected data are presented in Table 3. Based on them, it can be said that despite the subjective approach to the evaluation of individual methods of each person, the general qualitative trend is maintained quite well. Only methods 7 and 8 were assessed very similarly and it is difficult to clearly indicate a better one.

### 3.2. Analysis Using the RGB Model

The RGB model was used according to the procedure described in Section 2.2, based on the average scores given to the criteria from a pool of all 12 evaluators. Mean values of the scores and the resulting parameters are summarized in Table 4.

#### 3.2.1. Analysis of the Overall Potential

The position of a given method is best evidenced by the values of the mean overall assessment (whiteness), shown in Table 4, which expresses the same trend as the data presented earlier in Table 3. Of all the methods, three are clearly the best overall: FIA-SF, with a noticeable advantage over FIA-DAD and DPASV. Although two flow methods be-long to this “top class”, such methods (FIA-ICP/MS and SIA-CE/DAD) were also rated as definitely the worst. It is worth noting, however, that the difference between the overall scores of the best and worst methods is about 20%, which is a relatively small value taking into account the relatively large number of assessed methods, as well as their highly different specificity and characteristics in individual areas.

#### 3.2.2. Analytical Aspects

The results obtained for the three main attributes significantly differ from the trend discussed earlier, obtained for the overall assessment. The method of greatest analytical potential turned out to be FIA-ICP/MS (93.0%), which was assessed as one of the worst in the overall perspective. The second method in the “red” category was FIA-SF (89.8%), which in turn performed best overall. However, when taking into account the assessment of the individual red criteria, there are significant differences. For example, clearly different ratings were given for R2 (LOD and LOQ) and R4 (accuracy), which, interestingly, for these methods are inversely correlated, i.e., the one with the best accuracy (FAAS) shows the weakest LOD and vice versa. This is probably coincidence. The SP method has one very strong analytical side—R4 (accuracy), although there is a dissonance between its accuracy and precision (rated at 94% and 66%, respectively). The methods with the lowest analytical potential turned out to be FIA-DAD (65.3%) and SIA-CE/DAD (63.0%), which to some extent can be explained by the use of the same not-sophisticated type of detection—DAD. However, significant differences between these methods are observed for R2 (LOD and LOQ)—here the SIA-CE/DAD was rated much better, and R3 (precision)—here the FIA-DAD performs better. The reason for this is probably the specificity of the SIA-CE/DAD method, which uses a special module for concentration of the analyte by evaporation in the flow system, and on the other hand, separation using the CE technique, one of the biggest drawbacks of which is the poor repeatability of migration times due to the instability of electroosmotic flow.

#### 3.2.3. Ecological Aspects

The most environmentally friendly method is unambiguously the FIA-DAD method (90.9%). Each green criterion of this method has been assessed at over 80%, which means that it is in fact close to the ideal in terms of ecology and safety of use. Particular attention is paid to the high score compared to other methods in the case of the G2 criterion (waste), which thanks to the use of the flow module have been significantly minimized. The next greenest methods are DPASV (84.5%), SP (82.5%) and FIA-SF (80.5%). Interestingly, the result of the FIA-SF method would have been better if not for the G1 (toxicity) criterion, which was poorly assessed due to the long list of chemicals hazardous to the environment and health. The FAAS, ICP/OES and SIA-CE/DAD methods turned out to be less environmentally friendly, but still with greenness values above 70%. In the case of the FAAS and ICP/OES methods, the worst rated criterion was G3 (energy in-take), which is closely related to advanced and energy-consuming equipment. The weakest point of the SIA-CE/DAD method is G1 (toxicity). Its greenness could be increased by replacing the used reagents with safer and ecological equivalents. The least environmentally friendly method in this comparison is FIA-ICP/MS, for which three out of the four criteria were rated very low: G1 (toxicity), G2 (waste) and G3 (energy intake). The use of the flow module in this method did not significantly improve this attribute, as it was, however, for the FIA-DAD and FIA-SF methods. The reason is probably the highly complex nature of these methods, combining different experimental techniques. In addition, the use of the MS detector is always involved with a large consumption of electricity, which is related to the need to constantly operate the vacuum pumps, which are responsible for the consumption of the largest amount of electricity among all the detectors discussed here.

#### 3.2.4. Practical Aspects

As in the case of the green attribute, the FIA-DAD method scored the best in terms of the practical criteria (89.9%). B1 (costs) and B2 (speed) were rated particularly highly. The FIA-SF method (86.6%) turned out slightly worse. Other methods were significantly lower rated in this respect. Average assessment values above 70% were recorded for electrochemical techniques for which B2 (analysis time) was the worst-rated criterion. What draws attention, is their high score in the B4 criterion (ease of use), which is associated with a significant degree of miniaturization of the instrument and its relatively easy transportation to the place of target measurements (portability). The FAAS method (68.1%) was next, which, along with the FIA-DAD method, has the best rated speed of analyses. The other three methods were clearly assessed worse: ICP/OES (59.4%), SIA-CE/DAD (58.2%) and FIA-ICP/MS (46.8%). The ICP/OES method performs well in terms of B2 (speed), which is not surprising because it is related to the FAAS method, which was assessed excellently in this respect. On the other hand, it performs poorly in relation to criteria such as B1 (cost) and B4 (ease of use). Interestingly, the SIA-CE/DAD method turned out to be very cheap, but at the same time the slowest of all the methodologies discussed. This is due to the slow evaporative concentration step and the off-line coupling with CE.

#### 3.2.5. Overall View

Overall, the best method in the presented ranking is by far the FIA-SF, which wins with high scores for each attribute. The second method in the ranking, FIA-DAD, has better assessed ecological and practical criteria, but loses in the analytical aspect. While the worse LOD value seems obvious in this case, the greater accuracy could be achieved, for example, by a better selection of the calibration strategy and a method of eliminating potential interference effects [25,26].

The data in Table 5 are noteworthy, showing the differentiation of ratings between all 12 criteria for individual methods, and between 4 criteria of the respective attribute. The lowest RSD values characterize the most balanced and equilibrated methods in given aspect. From a global perspective (differentiation of all 12 criteria), the best are electrochemical methods (DPASV and SP), and only slightly worse spectroscopic methods integrated with the flow system (FIA-SF and FIA-DAD). This indicates the high versatility of these methods. As similarly assessed in terms of different criteria, they show a similar potential in relation to different expectations, e.g., both high-quality analyses and routine, performed on a large scale. The weakest in this respect is the FIA-ICP/MS method. Thus, the degree of comprehensive balancing seems to be inversely proportional to the technological advancement of the detection equipment and the analytical procedure itself. Table 4 also shows that the differentiation of scores in case of the analytical criteria is generally smaller than the remaining ones—for which the RSD values exceed 30% in some cases or even 40%. It is due to the fact that among the ecological and practical criteria one can find the bottlenecks of particular methods that strongly affect the overall assessment. This shows once again how important it is to take into account those attributes and consider them in the same manner as the analytical criteria. Consequently, one can state that being aware of the greatest limitations of a method can often be more important in its selection than knowing about its greatest advantages.

Another perspective is presented by the data in Table 6, which shows how well individual criteria were assessed as a whole, and what the variability was of the assessments of the given criteria resulting from the variety of the considered methods. It turns out that the best rated criteria, with average values above 80%, were: R3 (precision), R4 (accuracy) and G4 (direct impact). This is probably due to the fact that, as analysts, participating in the development of new methods, we are used to taking care, above all, of the quality of the analytical results determined by precision and accuracy. The slightly lower value of the mean score for the R2 criterion (LOD and LOQ) may result from a greater discrepancy of this parameter between the methods, which was involuntarily taken into account by the evaluators. The best rated G4 criterion takes into account additional hazards such as the risk of burns, exposure to poisonous vapors, cuts, working with high pressure gases, etc., as well as whether the methods use animals or genetically modified organisms. It is presented in total as a direct influence of the method on man and animated nature and its naturalness.

Criterion B4 (operational simplicity) was rated the worst of all, and this is related to the relatively low level of miniaturization, automation, integration, and portability of the considered methods. These methodologies are mostly intended to be used only in a laboratory with the constant assistance of qualified personnel. The second lowest rated criterion is G1 (toxicity), which to some extent may reflect a general lack of care for the characteristics of the reagents used. The greatest differentiation of assessments was noted for two practical criteria: B1 (costs), B2 (speed) and one ecological criterion—G1 (toxicity). Importantly, these criteria, due to the fact that they strongly differentiate the considered methods, were decisive for their position in the ranking. At the same time, they are the best field for potential improvements of individual methods in order to raise their overall assessment.

## 4. Conclusions

When comparing different methods and choosing the best one, it is important to determine what the primary criterion for selection is—the overall potential (whiteness), the score obtained for a given trait (redness, greenness, blueness), the evaluation of a specific criterion or several of them, or the degree of differentiation of individual criteria. It is worth realizing that only the analysis of all these parameters gives a complete set of information and a full picture of the strengths and weaknesses of a given method. From a practical point of view, it seems reasonable to define one or several of the most important criteria, bottlenecks of the method in relation to its planned use, and then to make a preliminary selection of methods from their point of view, e.g., by determining the minimum acceptable score. After such a pre-selection, the subsequent overall assessment by the whiteness method avoids the risk of choosing a “trap method”, and takes into account the various features of the method: quality of results, practical aspects, environmental performance, and thus is closest to the idea of sustainable development [27,28].

It is worth noting the correlation between the degree of technological advancement of equipment and the complexity of the methodology, and the disproportion between the evaluation of the red attribute and the others. ICP and MS are advanced technologies compared to DAD, which, although they allow for high quality results, are associated with high costs, difficult handling, high requirements and a general lack of environmental performance. This disproportion in favor of red aspects may show significant advantages for exceptional, control, sporadic, small-scale analyses, where environmental impact is therefore limited, even for poorly assessed greenness. Methods that are primarily green and blue are ideal candidates for large-scale screening methods, provided, of course, that the minimum requirements for key analytical aspects are met. 

To maximize the potential of flow techniques-including high analytical speed, automation of sample preparation, low analytical cost, low reagent consumption and low waste production—they should be combined with relatively simple detection techniques and reduce the complexity of the analytical procedure. The validity of this approach is demonstrated by the highly evaluated FIA-SF and FIA-DAD methods compared to the SIA-CE/DAD and FIA-ICP/MS methods. In addition to spectral detection systems, electrochemical detectors can exhibit adequate characteristics for integration with a flow-through format.

The approach to the evaluation of methods based on the RGB model presented here, based on the recruitment of a wide range of analysts with extensive experience, is not free from disadvantages. The problem of the objectivity of the assessment of individual criteria seems obvious. However, this effect is reduced to some extent by averaging the assessments of given criteria. An important limitation is also the lack of data allowing reliable assessing of certain criteria, which forces the necessity to estimate and generalize. Therefore, these results obtained should be treated critically, rather as an attempt to set a new direction in the evaluation of methods and obtain general information of an auxiliary nature relating to the specificity of given methods, rather than clearly indicating the superiority of one method over another.

Our recommendation to other researchers is to attempt evaluation of their own methods using the RGB model (the freely available online Excel spreadsheet template may also be helpful for performing method assessment [13]). The data provided may allow for the development of additional guidelines for the assessment of individual criteria in the future, and thus contribute to an increase in the objectivity of assessments and comparisons using this tool.

## Figures and Tables

**Table 1 molecules-26-03914-t001:** The analytical methods studied in this work.

Abbreviation, Measurement System *	Sample	Procedure	Ref.
SIA-CE/DAD, UV/VIS-DAD	Drinking and waste water	Flow module on line connected with a vaporizer to concentrate the sample and off-line with capillary electrophoresis to separate the sample components	[14]
FAAS, AAS	Waters	The ISO procedure experimentally verified by authors in terms of quantitative parameters using PinAAcle 900 AA spectrometer (Perkin Elmer Inc., USA)	[15]
ICP/OES, OES	Waters	The ISO procedure experimentally verified by authors in terms of quantitative parameters using Optima 2100 DV spectrometer (Perkin Elmer, Inc, USA)	[16]
SP, potentiometry	Lake and effluent water	Procedure based on simultaneous preconcentration and reduction of metal ions onto a multiwall carbon nanotube electrode followed by subsequent chemical stripping	[17]
DPASV, voltammetry	Lake water	Procedure based on the use of a disposable sensor—a screen-printed carbon electrode co-modified with an in situ plated bismuth film and gold nanoparticles	[18]
FIA-DAD, UV/VIS -DAD	River water	Procedure based on the use of PAR (4-(2-pyridylazo) resorcinol) as colorimetric reagent and multivariate calibration for the determination of Zn, Cu and Mn in river water samples	[19]
FIA-SF, spectrofluorimetry	Food	Procedure based on the fluorescence of the zinc-8-(benzenesulphonamido) quinoline chelate in a micellar medium of sodium dodecylsulfate	[20]
FIA-ICP/MS, MS	Ocean seawater	Preconcentration of metals using a column with chelating resin	[21]

* SIA-CE/DAD—sequential injection analysis-capillary electrophoresis with diode array detection; FAAS—flame atomic absorption spectrometry; ICP/OES—inductively coupled plasma/optical emission spectrometry; SP—stripping potentiometry; DPASV—differential pulse anodic stripping voltammetry; FIA-DAD—flow injection analysis with diode array detection; FIA-SF—flow injection analysis with spectrofluorimetric detection; FIA-ICP/MS—flow injection analysis-inductively coupled plasma mass spectrometry.

**Table 2 molecules-26-03914-t002:** Values of the key parameters used in the evaluation (taken from the literature or estimated).

Method	LOD (µg/L)	RSD (%)	Relative Error (%)	Total Number of Pictograms	Waste Production (mL/10 Samples) *	Occupational Hazards *	Estimated Cost (EUR) *	Estimated Speed of Analysis (s Per Sample) *
SIA-CE/DAD	25	6.0	7.8	14	190	3	20	60
FAAS **	30	2.8	3.0	4	225	4	50	1
ICP/OES **	3	3.0	5.0	4	230	4	250	2.5
SP	28	5.6	2.1	8	300	1	12	11
DPASV	0.05	2.8	15.0 ***	6	300	1	12	7.5
FIA-DAD	72	3.7	12.0 ***	4	110	0	9	1
FIA-SF	0.2	1.1	0.6	14	110	0	11	1.3
FIA-ICP/MS	0.001	3.0	1.0 ***	19	900	3	490	8.8

* mean of all estimates, ** data in Perkin Elmer manual [24], *** recalculated based on the recovery data (maximum error).

**Table 3 molecules-26-03914-t003:** Ranking of methods according to overall assessments (whiteness); 1 and 8: the best and the worst methods.

Method	Evaluating Person												Mean
	I	II	III	IV	V	VI	VII	VIII	IX	X	XI	XII	
FIA-SF	1	2	1	1	1	1	3	1	1	2	1	1	1.3
FIA-DAD	2	1	3	2	3	2	4	3	4	1	2	2	2.4
DPASV	6	3	2	3	2	3	1	5	2	3	3	4	3.1
SP	3	4	4	4	4	4	2	4	3	4	5	7	4.0
FAAS	3	5	5	5	7	6	5	2	5	5	4	3	4.6
ICP/OES	5	6	6	6	6	5	6	6	6	7	6	5	5.8
SIA-CE/DAD	7	7	7	7	8	8	7	7	7	6	8	8	7.3
FIA-ICP/MS	8	8	8	8	5	7	8	8	8	8	7	6	7.4

**Table 4 molecules-26-03914-t004:** Method evaluation results obtained using the RGB model.

Method	Aspects Evaluated (%)												Whiteness (%)
	Analytical (Red)				Ecological (Green)				Practical (Blue)				
	R1	R2	R3	R4	G1	G2	G3	G4	B1	B2	B3	B4	
FIA-SF	66.7	94.6	98.3	99.6	46.7	92.1	83.6	99.7	92.9	97.8	86.0	69.9	85.7 **
	89.8 *				80.5 *				86.6 *				
FIA-DAD	60.4	50.0	82.5	68.3	90.4	90.4	83.2	99.7	97.9	100.4	86.5	75.0	82.1 **
	65.3 *				90.9 *				89.9 *				
DPASV	75.4	99.2	91.3	66.7	79.2	71.7	89.5	97.4	89.2	67.9	73.3	79.4	81.7 **
	83.1 *				84.5 *				77.5 *				
SP	69.2	64.2	66.3	94.1	73.8	69.2	89.5	97.4	89.2	60.8	73.5	79.3	77.2 **
	73.4 *				82.5 *				75.7 *				
FAAS	77.5	63.2	91.3	86.3	92.1	69.4	63.2	87.0	62.1	100.4	69.2	40.7	75.2 **
	79.5 *				77.9*				68.1 *				
ICP/OES	80.4	81.7	88.8	73.8	92.1	69.6	55.5	87.4	40.8	88.8	68.1	40.0	72.2 **
	81.1 *				76.1 *				59.4 *				
FIA-ICP/MS	82.1	103.3	89.2	97.5	35.0	42.9	50.0	90.2	25.8	65.3	52.5	43.5	64.8 **
	93.0 *				54.5 *				46.8 *				
SIA-CE/DAD	59.2	67.7	64.6	60.4	46.7	82.5	67.7	91.2	81.3	26.3	64.5	60.7	64.4 **
	63.0 *				72.0 *				58.2 *				

The numbers represent the scores awarded to the particular method and criterion. *—mean for four criteria of the given color, **—mean for all 12 criteria.

**Table 5 molecules-26-03914-t005:** The variance of assessments expressed as RSD (%) of the main assessment outcomes, presented for the individual methods.

Method	Aspects Studied			
	Analytical (*n* = 4)	Ecological (*n* = 4)	Practical (*n* = 4)	Overall (*n* = 12)
SIA-CE/DAD	6.2	27.1	39.7	26.4
FAAS	15.5	17.7	36.3	22.4
ICP/OES	7.6	22.1	39.6	25.3
SP	19.0	16.1	15.6	16.0
DPASV	17.8	13.4	11.8	13.8
FIA-DAD	21.0	7.4	13.0	19.4
FIA-SF	17.3	29.2	14.1	19.3
FIA-ICP/MS	10.0	45.0	35.4	40.9

**Table 6 molecules-26-03914-t006:** The average score for each criterion with the RSD value (%) calculated for all eight methods.

Criterion	Mean	RSD (*n* = 8)
R1: Scope of application	71.4	12.4
R2: LOD and LOQ	78.0	25.1
R3: Precision	84.0	14.6
R4: Accuracy	80.8	19.0
G1: Toxicity of reagents	69.5	33.5
G2: Amount of reagents and waste	73.5	21.2
G3: Energy and other media	72.7	20.4
G4: Direct impacts	93.8	5.7
B1: Cost-efficiency	72.4	36.9
B2: Time-efficiency	75.9	34.2
B3: Requirements	71.7	15.5
B4: Operational simplicity	61.1	28.4

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
