# Peer review of "Comprehensive Assessment of Flow and Other Analytical Methods Dedicated to the Determination of Zinc in Water"

_molecules, 2021, doi:10.3390/molecules26133914_

Round 1

Reviewer 1 Report

General comment:

The present manuscript deals with the application of a new methodology used to evaluate and classify analytical methods. The authors have previously published this methodology and this work represents the application to a particular case, i.e., the comparison of methods for the determination of zinc in water.

The manuscript is very well written, but I would like to point out some issues that I have found.

In lines 124-141 the authors claim that “It is also possible to evaluate individual criteria based on predefined guidelines, but in this case such additional standards were not applied”. In my opinion, this almost invalidates the results of the comparison since it all depends on the criteria of singular people, rather than only on numbers and figures. This makes the comparison too subjective and not so objective. This is also observed in lines 150-159 where the authors claim that “Where relevant data were not available, the missing values were subjectively estimated based on best evaluator's knowledge and intuition.” Again, in my opinion, this makes the results too subjective. In addition, only 12 people with unknown backgrounds (for confidentiality or anonymity reasons, I understand) are responsible for this work. I am not sure  if this is statistically valid enough to make conclusions. Results shown in table 4 are in agreement with table 3 but this is only because both tables are representing the same information in different ways. The final “whiteness” results are statistical means obtained from data provided by the 12 people participating in this work. I am not sure if using parametric statistics is the best way to evaluate the information, since, if one or two of the participants give an extreme value to one of the RGB results, this would affect the mean in excess. In my opinion, it would be necessary to increase the number of specialists evaluating the works and only use objectively obtained figures present in the manuscripts evaluated, rather than “subjectively estimate” any of them.

Finally, the authors say in the Conclusions that “these results should be treated critically, rather as an attempt to set a new direction in the evaluation of methods and obtain general information of an auxiliary nature relating to the specificity of given methods, rather than clearly indicating the superiority of one method over another”. It seems that after all the effort made to compare the different methods, this evaluation methodology does not allow to objectively choose a particular method as “the best”, but only serves as a mere comparison, which many authors are already adding in their manuscripts in sections such as “Comparison with other methods” in a much easier way.

Although the idea seems brilliant, in my opinion it is still in a very early stage, and it is not easy to compare methods until a very specific guide can be applied. Only when no “subjectively estimated” numbers are used, these types of works could be published. I would recommend the authors to remove elements such as “toxicity”, “impact”, or “simplicity”, which seem too general, subjective, and difficult to estimate numerically.

Finally, I would like to point some minor issues I have found in the manuscript that might help the authors:

Introduction:

  • Lines 58-59: change “The model assigns three primary colors - red, green and blue - three different attributes of the method” to “The model assigns three primary colors - red, green and blue – to three different attributes of the method”.

Results and discussion:

  • Line 233: add the percentage symbol (%) after figure “80”.
  • Lines 299: change “analyzes” to “analyses”

Tables:

  • Table 1: remove the “commas” in the “procedure” column in lines of methods of “SP” and “DPASV”. I would also suggest adding the meaning of the abbreviations at the table foot.
  • Table 3: remove this table. In my opinion it is not giving any extra information more than table 4 gives.
  • Table 4: indicate that below the four RGB values given for each method, the mean is given. As it is now, it looks like a fifth value. Also, add the percentage symbol (%) after “mean whiteness”.

Reviewer 2 Report

Dear authors

Even though it was indicated that the classification of 12 experienced people was used, it is not possible to quantify the degree of involvement/expertise, that is, we do not have the characterisation of these people and so the study seems to me very inconsistent, not to mention the subjectivity of the classification, even using a scale. It seems to me more like a sociological study than an analytical methodology. The study requires more participants and more values and little or no estimates.
The authors themselves indicate in their conclusion the inconsistency of their results.

Best regards

Reviewer 3 Report

The MS refers to a very important theme, is original and can be regarded as an amalgamation of relevant recent concepts. Its publication is therefore recommended.

In order to contribute to improve the MS quality, this referee took the liberty of stressing some aspects for the authors reflection, and highlighting some required text corrections and/or modifications.

General aspects:

Text conciseness is lacking, with plenty of too long sentences and some redundancy.

A text on novelty statement, perhaps at the end of the introductory section, is missing.

FIA and SIA acronyms could be replaced simply by FA (Flow Analysis), as suggested in Anal. Chim. Acta, 2020, 1093, 75-85.

The meanings of words “method”, “technique” and “procedure” are sometimes overlapped.

Comments on “robustness” (mentioned once in page 4) and “ruggedness” (not mentioned) are lacking.

Specific aspects

Title - the expression “non-flow” appears only once in the abstract and never in the main text; the word “comprehensive” is not clear unless the entire manuscript is read.

Abstract and keywords - a modified version could perhaps be:

“An original strategy to evaluate analytical procedures is proposed and applied to verify if the flow-based methods, generally favorable in terms of green chemistry, are competitive when their evaluation relies also on other criteria. To this end, eight methods for the determination of zinc in waters, including four flow-based ones, were compared and the RGB model was exploited. This model takes into account several features related to the general quality of an analytical method, namely its analytical efficiency, compliance with the green analytical chemistry, as well as practical and economic usefulness. Amongst the investigated methods, the best was the flow-based spectrofluorimetric one, and a negative example was that one involving a flow module, ICP ionization and MS detection, which was very good in analytical terms, but worse in relation to other respects, which significantly limits its overall potential. Good assessments were also noted for non-flow electrochemical methods, which attract attention with a high degree of balance of features and, therefore, high versatility. The original attempt to confront several worldwide accepted analytical strategies, although to some extent subjective and with limitations, provides interesting information and indications, establishing a novel direction towards the development and evaluation of analytical methods.

Keywords - flow analysis; green analytical chemistry; method assessment; RGB model; validation; zinc determination in waters”.

Introduction – lines 42-44: check grammar. Perhaps remove authors names? Add et al after Armenta and after Galuszka?).

line 73: remove “above 5 mg/L”.

third paragraph: delete NEMI, GAPI, AGREE, as they are not used along the MS.

Lines 48-52: edit to “By taking into account the above guidelines, several tools for assessing the greenness of a method such as National Environmental Methods Index [5], Green Analytical Procedure Index [6], Eco-Scale and Analytical Greenness Calculator [8] have been recently proposed. These tools allow one to graphically express the greenness of a method…”.

Lines 56-66 (here and elsewhere): the three fundamental colors (Newton disk) are red, yellow and blue (RYB) and not red, green and blue (RGB), yet this later acronym became popular after the advent of the light emitting diodes. This referee does not know if the entire MS text should be amended in this regard.

Line 74: instead of “desired”, write “desirable”.

Ln 77: delete “such”.

Materials and methods – instrumental operating conditions are missing.

Table 1 – columns 1 and 3 (abbreviation and measurement system): combine in one; column 2: delete; column 5: remove verb [ex: Flow module on line connected …; ISO procedure experimentally verified…, etc], line 7: edit to “Spectrofluorimetry” and “Procedure”; SP acronym is awkward.

Line 100: instead of “refers to 12 general rules” write “refers to the 12 general rules [4]”.

Lines 124 – 128: vague text.

Table 2 - column 1: methods acronyms should be improved according to Table 2 (after combining column names) - idem for the other tables;

Line 146: edit to “*mean of all estimates, **data in PerkinElmer manual [25]…”

Line 149: find a better expression to replace “Methodology for the evaluation of methods”.

Line 171: instead of “…to indicate a clearly better method”, write “to clearly indicate the better one”.

Table 3 - edit caption to “Ranking of methods according to overall assessments (whiteness); 1 and 8:  the best and the worst methods”.

Line 178: the entire 3.2 text from here to line 348 should be amended, reduced and simplified, in order to avoid repetitive phrases, redundancies and repeated information.

Line 180: delete “All the”.

Table 4, first line, top: delete “by the RGB model”; three central columns: define numbers (as foot notes) at the bottom of each cell.

Line 187: delete “shown”.

Line 188: Instead of “Of…” write “Among…”.

Line 191: delete “definitely”.

Line 199: “Analysis of analytical…” sounds strange. What about to remove “analysis of” from 3.2.1, 3.2.2, 3.2.3, 3.2.4 captions?

3.2.2 item (here and elsewhere): “analytical potential” is vague. Better to define this expression when it appears from the first time.

Line 315: “variability of assessments” is vague; the n-value should be defined as food note.

Line 350: “4. Conclusions” too long a text. It could perhaps be replaced by “4. Final remarks” and the related text modified accordingly. Thereafter, a short finalizing text (one - two small paragraphs) could be added as item “5. Conclusions”. Of course, this depends on the manuscript requirements inherent to Molecules.

References 5, 6, 8, 15 and 28: article titles must be written in lowercase, exception for the first one.

Refs 14 and 17 refer to recommended methods. Would it be plausible to replace them?

Ref. 21: edit authors names to “Tarley, C.R.T.; Santos V.S.; Baeta, B.E.L.; Pereira, A.C.; Kubora, L.T.” Check please!

Ref 23: instead of “Filho, O.F.”, write “Fatibello-Filho, O.”. Note: “Filho” (in Portuguese) means “Junior” (in English)”.

Round 2

Reviewer 1 Report

The authors have made all the most important proposed changes to the manuscript. In addition, it is clearer now what the novelty of this work is, and that one should be very careful when using the RGB model.

Although the authors quite explain that the results are somehow subjective and that they have tried to have a larger data set, this might stand for a first attempt of contributing to this way of comparing different methods.

Reviewer 2 Report

Dear authors

The work requires greater consistency in the results and detailed characterisation of the participants, without jeopardising anonymity. I do not recommend the publication of this work before these weaknesses are resolved, even though the authors themselves have indicated the inconsistency of their results.

Best regards